# Pro-Gastrin-Releasing Peptide as a Biomarker in Lung Neuroendocrine Neoplasm

**DOI:** 10.3390/cancers15133282

**Published:** 2023-06-22

**Authors:** Violetta Rosiek, Angelika Kogut, Beata Kos-Kudła

**Affiliations:** 1Department of Endocrinology and Neuroendocrine Tumours, Department of Pathophysiology and Endocrinology, Medical University of Silesia, 40-014 Katowice, Poland; 2Department of Endocrinology and Neuroendocrine Tumours, Medical University of Silesia, 40-014 Katowice, Poland

**Keywords:** biomarker, pro-gastrin-releasing peptide (ProGRP), chromogranin A (CgA), neuroendocrine neoplasm (NEN), lung

## Abstract

**Simple Summary:**

The aim of this study was to evaluate the diagnostic effectiveness of pro-gastrin-releasing peptide (ProGRP) and chromogranin A (CgA) in the diagnosis of neuroendocrine neoplasms of the lung (LNENs), (290 cases) and compared these results with controls (54 cases). The median ProGRP levels in LNEN patients were higher compared to controls (136.4 pg/mL vs. 6.5 pg/mL). The majority of the LNEN was well-differentiated tumors (262) (typical and atypical carcinoid). Based on the results ( sensitivity, specificity, and area under the curve) of ProGRP in LNENs vs. controls, we can conclude that ProGRP should be considered as an effective marker for the diagnosis of LNEN patients.

**Abstract:**

There is a lack of effective biomarkers for diagnosing lung neuroendocrine neoplasms (LNENs). A known small cell lung cancer (SCLC) biomarker is a pro-gastrin-releasing peptide (ProGRP), but not for all LNENs, especially for bronchopulmonary carcinoids. This study aimed to evaluate the diagnostic value of ProGRP and chromogranin A (CgA) in diagnosing LNENs. The ProGRP and CgA levels in 290 cases of LNENs and 54 healthy controls (HCs) were measured. The median ProGRP concentration in the group of LNEN patients was 136.4 pg/mL, higher than that of HCs at 6.5 pg/mL. Most of the LNEN cohort was well-differentiated tumors (typical and atypical carcinoids, *n* = 262, 91.7% of all LNENs). The sensitivity, specificity, and area under the curve (AUC) of ProGRP when distinguishing LNENs vs. HCs were 94.8%, 100%, and 0.995. CgA (AUC = 0.375) could not determine LNENs vs. HCs. Therefore, based on these results, ProGRP may be considered as an effective marker for diagnosing LNENs.

## 1. Introduction

Primary lung neoplasia is one of the most common and fatal groups of malignant tumors, with a 5-year survival rate of about 15% [1] and still higher mortality [2,3]. Among them, 25% account for the neuroendocrine neoplasms (LNENs), which are relatively rare tumors, and regarded as 20–25% of all neuroendocrine neoplasms (NENs) [4]. According to the World Health Organization (WHO) 2015 Classification, LNENs include several neoplasias, such as typical carcinoid (TC), atypical carcinoid (AC), large cell neuroendocrine cancer (LCNEC), and small cell lung cancer (SCLC) [5]. The Ki-67 index, histopathological aspect, mitotic rate, and presence of necrosis are the criteria on which this classification is based. This histological differentiation is crucial in choosing the correct therapeutic methods and determining the prognosis. Immunohistochemical examinations for neuroendocrine markers in LNENs like chromogranin A, neuron-specific enolase (NSE), CD56, and synaptophysin can differentiate NENs [6]. The best-known biochemical markers used in the diagnosis of lung cancer are carcinoembryonic antigen (CEA), cytokeratin 19 fragment (CYFRA21-1), and pro-gastrin-releasing peptide (ProGRP), the use of which is becoming more widespread [7].

ProGRP has been described as a tumor marker for SCLC [8,9]. ProGRP is a precursor form of gastrin-releasing peptide (GRP), which is a neuropeptide hormone originating from the porcine gastric [10], and is widely distributed in the gastrointestinal and pulmonary system [11]. ProGRP, compared to GRP, is very stable and has a long half-life (19–28 days vs. 62 min). ProGRP has been studied extensively in SCLC [12,13]. However, very little data are available on ProGRP levels in the blood serum of patients with LNENs [14,15,16,17,18,19,20].

Chromogranin A (CgA) is the most often used tumor marker in NENs of lung origin. According to the latest recommendations of the European Neuroendocrine Tumor Society (ENETS), CgA can help diagnose and monitor the course of the disease and detect progression and recurrence [21]. However, CgA has limitations because elevated CgA levels can occur in chronic atrophic gastritis, inflammatory diseases, renal/liver impairment, and during proton pump inhibitor treatment [21]. 

The improvement of diagnostic methods, including new, more sensitive biomarkers, would help speed up LNEN detection and allow the initiation of treatment in the earliest stage, which would significantly reduce mortality and improve prognosis. Therefore, we tried to evaluate the diagnostic utility of a ProGRP assay in serum for LNENs, and we compared its accuracy to that of CgA to better identify the likelihood of LNENs. We previously analyzed other serum markers in LNEN patients and the control group, but these results were not excellent for distinguishing these groups [22].

Unfortunately, our studied and analyzed group did not include SCLC. These patients were not admitted to the endocrinologist (to our Department of Endocrinology and Neuroendocrine Tumors), but in the first choice, they were referred to the oncologist according to the poorly differentiated maturity of the tumor (to the Department of Clinical Oncology in our hospital).

## 2. Materials and Methods

### 2.1. Strategy

We compared serum ProGRP and CgA levels from LNENs to healthy controls. We calculated both groups’ diagnostic accuracy and metrics (AUROC, sensitivity, and specificity) for ProGRP and CgA. The study was conducted at the Department of Endocrinology and Neuroendocrine Tumors, European Neuroendocrine Tumor Society Centre of Excellence, Medical University of Silesia, in accordance with the good clinical practice guidelines and the Declaration of Helsinki.

### 2.2. Cohorts

All of the study subjects provided written informed consent for blood analysis, authorized by the institutional (Medical University of Silesia) ethics committee. We conducted a retrospective, one-center cohort study between April 2020 and August 2022 of LNEN patients and nonaffected patients attending the occupational medicine clinic. LNEN patients with histological confirmation of the disease met the criteria for inclusion. The following exclusion criteria were applied: coexistence of another malignant tumor, kidney and liver failure, current inflammation, and no written informed consent. The studied cohort included healthy controls (*n* = 54) and patients with LNEN (*n* = 290) of various disease states, grades, and histology (Table 1). The patient group comprised 192 typical carcinoids (62%), 74 atypical carcinoids (26%), and 22 large cell neuroendocrine carcinomas (8%). Additionally, two subjects had diffuse idiopathic pulmonary neuroendocrine cell hyperplasia as an incidental histological finding after the surgical removal of a lung tumor (Table 1). No small cell lung cancer cases were included because these subjects were not referred to us but to an oncologist according to the poorly differentiated maturity of the tumor. Control individuals were known to have an absence of malignancy and were asymptomatic and in good health.

### 2.3. Blood for ProGRP and CgA Measurement

Peripheral blood samples (5 mL) were collected during the first admission to the Department of Endocrinology and Neuroendocrine Tumors, placed in serum separator tubes with a clot activator, and allowed to clot for two hours at room temperature before centrifugation for 20 min at approximately 1000× *g*. Then, serum samples in aliquots were frozen and stored at −80 °C for later use, until the ProGRP and CgA assay measurements in the local laboratory of Silesian Medical University.

### 2.4. Histological Diagnosis

All LNEN patients in this study had histologically confirmed LNEN disease, described by an independent CoE expert pathologist according to the standard ENETS criteria and WHO 2015 classification. The same pathologist reviewed and evaluated all specimens (H and E, immunohistochemistry).

### 2.5. Biomarker Measurement

#### 2.5.1. ProGRP Measurement

Serum concentrations of ProGRP were measured using a pro-gastrin-releasing peptide (ProGRP) ELISA Kit, CEB186Hu (Cloud-Clone). According to the manufacturer’s instructions, the detection limit for ProGRP was <5.13 pg/mL, and the detection range was 12.35–1000 pg/mL. The inter- and intra-assay CV for ProGRP was <12% and <10%, respectively.

#### 2.5.2. CgA Measurement

Enzyme immunoassay (ELISA) was used to quantitatively determine Chromogranin A in serum (Demeditec Diagnostics GmbH, Kiel, Germany). The measuring range was 2.3–900 µg/L. A cut-off of 100 µg/L defined the upper limit of normal. The typical pathological range was up to 143,500 µg/L.

### 2.6. Radiological Evaluation of LNEN Disease

Disease extension was evaluated using anatomical imaging; either computed tomography (CT) or magnetic resonance (MR), as well as functional; [^68^Ga]Ga-DOTATATE PET/CT in well-differentiated LNENs (TC and AC) or [^18^F]F-FDG PET/CT in poorly differentiated LNENs (LCNEC).

### 2.7. Statistical Analysis

Inter-group analyses were undertaken using two-tailed nonparametric tests (the Mann–Whitney U-test for two groups or the Kruskal–Wallis test for multiple samples). Receiver operating characteristic (ROC) curve analysis was conducted for both serum markers (ProGRP and CgA) to understand their ability to discriminate LNEN patients from controls, leading to estimates of the area under the curve (AUC) with a 95% confidence interval (CI). An AUC value of >0.9 was considered to indicate an excellent diagnostic marker; values of 0.8–0.9 were considered good; values of <0.8 were deemed fair; and finally, values of <0.7 were poor. The calculated metrics included diagnostic accuracy, sensitivity, and specificity of ProGRP and CgA [23,24], and the Youden J index (performance of a diagnostic). Statistical analyses were performed using STATISTICA software (version 13.36.0, StatSoft, Poland). Statistical significance was defined at a value of *p* ≤ 0.05. Data are presented as the mean ± standard deviation (median). 

## 3. Results

### 3.1. Comparisons between ProGRP and CgA as a Diagnostic Test

The LNEN cohort comprised 266 bronchopulmonary carcinoid (BC) subjects (192 with typical carcinoids (TCs) and 74 with atypical carcinoids (ACs) and 22 LCNECs. Subjects with diffuse idiopathic pulmonary neuroendocrine cell hyperplasia (DIPNECH) (n = 2) were also included. In total, 72% of patients with carcinoids were stable at the time of blood draw (192/266); 74 (28%) had a progressive disease. A minority of these patients were metastatic (30.7%; regional 9.7%, distant 21%). A total of 18% of the patients with BC had disseminated disease (TNM IV).

We first evaluated serum ProGRP in the LNEN and control groups. ProGRP was significantly increased in LNEN patients (171.8 ± 156.2 [136.4:93.9–204.5]) compared to controls (8.3 ± 5.7 [6.5:5.5–8], *p* < 0.0001) (Table 2).

### 3.2. Relationship to LNEN Detection

#### CgA and ProGRP for Disease Detection

CgA-positives were detected in 46/290 subjects (29%). The Mann–Whitney U-test z statistic = −2.92, *p* = 0.0035 (Table 2).

ProGRP-positives were detected in 275/290 subjects (95%). The Mann–Whitney U-test z statistic = 8.20, *p* < 0.0001 (Table 2).

### 3.3. Diagnostic Accuracy of the ProGRP vs. CgA Assays

Diagnostic accuracy in the LNEN (TC + AC + LCNEC) cohort was 29% for CgA, compared to 95% for ProGRP. The metrics are included in Table 3 and Table 4. The ProGRP assay was significantly more accurate than CgA (χ^2^ = 32.2, *p* < 0.0001).

The ProGRP AUROC analysis value was 0.995 ± 0.03 (Figure 1). The z-statistic was 177.3, and the Youden J index was 94.8%. The accuracy, sensitivity, and specificity at the ProGRP cutoff of 32.3 were: 95.2, 94.8, and 100%.

## 4. Discussion

Neuroendocrine neoplasms of the lungs differ in their malignancy potential depending on the histological type. Typical carcinoids are characterized by low malignancy potential and atypical carcinoids by medium potential, while small cell carcinomas and large cell neuroendocrine carcinomas have a high malignancy potential [25,26]. It is of clinical significance to rapidly and appropriately diagnose these patients because this determines the start of treatment and the patient’s prognosis.

Serum markers of LNEN need to be well-recognized as factors in diagnosis, management, follow-up, and therapy monitoring. Unfortunately, only a minority of LNEN cases present with hormonal-related disorders, such as carcinoid syndrome, Cushing’s syndrome, acromegaly, or SIADH (inappropriate antidiuretic hormone secretion). A critical issue in diagnosing LNENs is the lack of an effective serum marker. There is currently promising, but not yet well-established, liquid biopsy techniques to detect neuroendocrine cells.

In LNEN disease, the most common secretory markers, such as serotonin, adrenocorticotropic hormone (ACTH), and growth hormone (GH), have proven to be valuable biochemical tools for specific clinical syndromes. However, a promising LNEN biomarker should capture both functional and nonfunctional tumors. The main limitation of LNEN diagnosis is that only poor practical blood markers exist. CgA is a nonspecific marker widely used in diagnosing and monitoring patients with all NENs. Its diagnostic value is limited by its low sensitivity and specificity. These limitations in identifying LNENs call for novel markers to be developed that have potential for clinical management.

Given the problems with CgA assessment results, we evaluated the effectiveness of CgA and another biomarker (ProGRP) in LNEN diagnosis. We compared two assays (CgA and ProGRP) performed in two independent laboratories, a laboratory of Silesian Medical University (ProGRP, ELISA assay) and a general hospital laboratory at the ENETS Center of Excellence (CgA, DD assay), to assess the diagnostic accuracy and clinical utility of the ProGRP assay to detect LNEN compared to CgA. 

This study was performed in a large LNEN cohort (n = 290) comprising TCs (n = 192), ACs (n = 74), and LCNEC (n = 22) to ensure accurate analysis. All samples were de-identified, and both assays were evaluated using biological material obtained at the same blood draw. Indeed, although the vast majority of LNENs were well-differentiated (TC and AC) lesions, about 9% had lymph node metastases, and 16% had liver metastases at the time of diagnosis.

The use of CgA is widespread in NEN detection, but in recent papers there has been growing recognition that CgA levels are not clinically useful in diagnosing LNENs [27,28,29]. Additionally, CgA has significant clinical limitations [30]. In a 2020 paper, Matar and co-researchers [29] showed that CgA is not a clinically useful biomarker in diagnosing lung NENs. Secretion is not considered an LNEN hallmark. Thus, many NEN assays which measure CgA do not define the neoplasm.

In our study, the concentration of CgA was significantly higher in the group of patients with LNENs compared to the control group (156.97 ug/L vs. 70.34 ug/L); however, the assessment of the diagnostic value of this marker in AUROC analysis was poor. The CgA assay was positive in 46 individuals (of 290 LNENs), and the diagnostic accuracy was only 29%. In contrast, ProGRP was positive in 275 subjects, and its accuracy was 95% (Table 3). In the study by Korse et al. of ProGRP, the ROC curve indicated a cut-off level of 90 ng/L with a specificity of 99% and a sensitivity of 43% in distinguishing lung tumors from other sites [15].

Because the CgA concentration is normal in 71% of LNEN cases, it is not surprising that it performed poorly in this study. Our CgA measurements failed to reflect LNEN diagnosis. The optimal metrics proposed for NEN biomarkers have been recommended to exceed 80% for sensitivity. Compared to the 16% sensitivity of the CgA assays, the ProGRP sensitivity was 95%, and the ProGRP test accurately identified LNEN cases. The ProGRP accurately (95%) differentiated LNENs from controls when a 32.3 pg/mL cut-off was used, whereas CgA detected only 29% of these cases.

In a 2016 Russian study, the authors showed that the highest concentration of CgA was found in patients with neuroendocrine neoplasms of the small intestine, colon, and pancreas [31].

ProGRP is also not a specific marker, as other studies have reported the usefulness of determining the concentration of this marker in other types of cancer as well, including medullary thyroid cancer (MTC). A 2021 study by Italian and Swiss researchers [32] showed significantly higher concentrations in patients with MTC (the mean concentration was 880 pg/mL) compared to patients after treatment of MTC (74.8 pg/mL) and patients with thyroid diseases other than MTC (46.3 pg/mL). They demonstrated diagnostic usefulness of ProGRP and monitored the effectiveness of the treatment of MTC using this marker. In our study, the mean ProGRP concentration for patients with LNENs was 171.8 pg/mL, while in the healthy control group it was 8.3 pg/mL. This was consistent with previously reported data where the mean ProGRP was increased in 50% of patients with evidence of disease, but with normal CgA [33].

ProGRP exhibited higher sensitivity (95%) than CgA (16%) in LNEN diagnosis (detection) (Table 4). The area under the curve (AUC) comparison of ProGRP (0.995) vs. CgA (0.375) also indicated LNEN diagnosis superiority. AUC values between 0.9 and 1 indicate an excellent biomarker (ProGRP), and values less than 0.6 indicate a failed biomarker (CgA).

In a heterogeneous LNEN group, based on local histopathologist expertise, this study indicated that the ProGRP test detected almost all LNEN cases, identifying LNEN with >90% accuracy. Based on this, using a cut-off of >32.3 pg/mL, there was a 95% probability that the tumor of the lung was a neuroendocrine neoplasia. 

There have been many promising papers on using circulating ProGRP as a diagnostic and prognostic marker for SCLC. However, studies investigating its usefulness in other histological types of LNENs, such as TC, AC, and LCNEC, are limited [18,20,34]. 

In a Chinese study from 2021 [35], researchers found that the concentration of ProGRP was significantly higher in patients with lung cancer compared to patients with benign tumors and a healthy control group. The same study showed that the concentration of this marker increased depending on the stage of the disease according to the TNM classification (the concentration for stage I patients was 51.06 pg/mL, while for stage IV patients, it was 89.42 pg/mL). The usefulness of ProGRP as a marker for differentiating SCLC from non-SCLC has also been proven: the serum concentration of ProGRP in patients with SCLC was significantly higher compared to its concentration in patients with squamous cell carcinoma and adenocarcinoma (85.63 pg/mL vs. 65.48 pg/mL vs. 70.69 pg/mL).

An Italian study from 2021 [36] obtained similar results: higher levels of ProGRP were found in patients with SCLC (1484 pg/mL) compared to healthy subjects (36.1 pg/mL) and patients with non-SCLC (45 pg/mL).

In recent years, scientific papers have reported on ProGRP as an optimal biomarker.

Niesman and co-workers showed in their study from 2023 that ProGRP has a high sensitivity (92.3%) in lung carcinoi with diffuse idiopathic pulmonary neuroendocrine cell hyperplasia. In addition, they proved that ProGRP was superior to CgA in diagnosis for proliferation, grading, staging, coexistence of diffuse idiopathic neuroendocrine cell hyperplasia of the lung, and response to treatment [37].

A study from 2023 from China reported that ProGRP levels in patients with SCLC were higher than in the healthy control group (*p* < 0.05) and in the benign lung disease group (*p* < 0.05); the sensitivity and specificity of ProGRP was estimated at 77.45% and 86.67%, respectively [38]. 

Another Chinese study in 2022 found that ProGRP has stronger diagnostic advantages than CEA and NSE in distinguishing SCLC from NSCLC [8].

Our study demonstrated that serum ProGRP can be used to accurately differentiate LNEN from HC with excellent metrics (an AUC of 0.995). Indeed, elevated levels of ProGRP in patient sera may help confirm the diagnosis of LNEN in cases of a diagnostic dilemma with clinical suspicion. However, our study had some limitations, including the small number of patients in the LCNEC subgroups (22/290) and the large difference in the number of LNEN patients and controls. Nevertheless, this study of the utility of ProGRP in lung neuroendocrine neoplasm diagnosis confirmed that ProGRP is accurate and specific for LNENs. This biomarker may represent an appropriate method to diagnose LNENs and facilitate the assessment of disease status. 

## 5. Conclusions and Limitations

Overall, our study demonstrated that a ProGRP assay in serum is significantly more effective than the clinically approved assay of CgA. This head-to-head comparison of the two serum assays reflected the superiority of the ProGRP test. Based on our study, the ProGRP test could be included in the diagnostic approach for LNEN patients.

The primary study limitation was the heterogeneity of LNEN patients (various disease states, grades, and histology) and the different numbers of LNEN patients and controls.

## Figures and Tables

**Figure 1 cancers-15-03282-f001:**
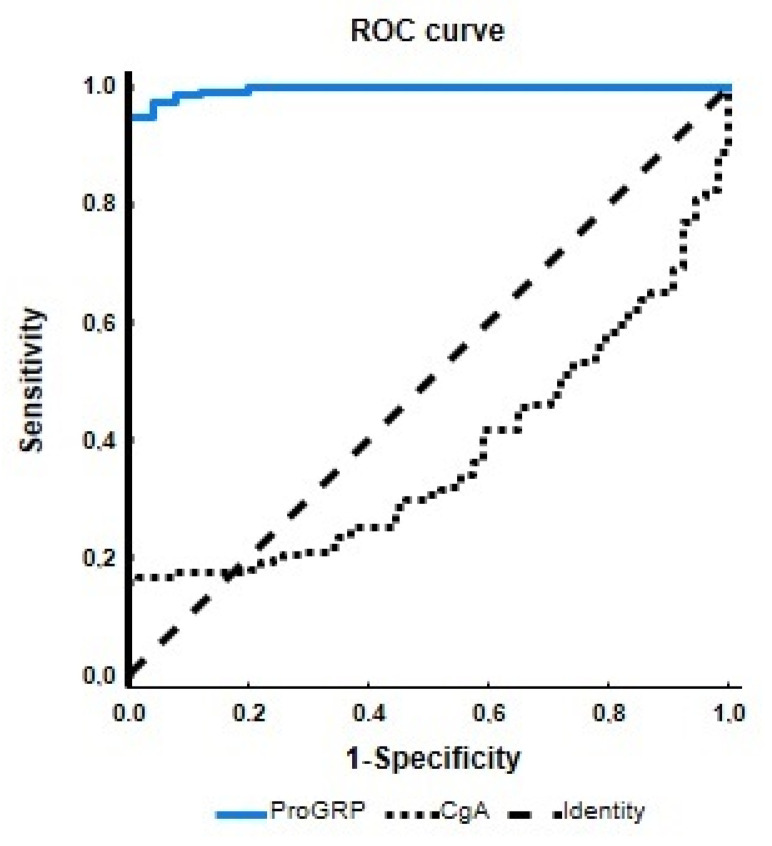
Performance of serum ProGRP and CgA for detecting lung neuroendocrine neoplasm (LNEN) patients. The receiver operating characteristic (ROC) curves and the area under the curves (AUC) for LNEN versus healthy controls are displayed. Individual ROC curves and AUC for serum ProGRP (AUC 0.995, 95% CI 0.990–1.000) and CgA (AUC 0.375, 95% CI 0.307–0.443).

**Table 1 cancers-15-03282-t001:** Demographic and clinicopathological characteristics of the study cohorts.

Variable	Category	LNEN Patients (*n* = 290)	Controls (*n* = 54)
ProGRP [N: 46 pg/mL]	Mean:Median	171.78:136.40	8.30:6.50
CgA [N: <100 ug/L]	Mean:Median	156.97:43.43	70.34:67.50
Age (years)	Mean:Median	58.74:62.36	39.84:38.00
Gender	Male:Female	80:210	18:36
Functional status	NF	270 (93.1%)	N/A
	F (CS)	19 (6.6%)	
	F (CD)	1 (0.3%)	
Disease status	SD	201 (69.3%)	N/A
	PD	89 (30.7%)	
TNM stage	Localized	201 (69.3%)	N/A
	Regional metastatic	28 (9.7%)	
	Distant metastatic	61 (21.0%)	
Histology	TC	192 (66.2%)	N/A
	AC	74 (25.5%)	
	LCNEC	22 (7.6%)	
	DIPNECH	2 (0.7%)	

Data are shown as mean and median or as numbers and percentages (%). Abbreviations: LNEN, lung neuroendocrine neoplasm; AC, atypical carcinoid tumor; LCNEC, large cell neuroendocrine carcinoma; TC, typical carcinoid tumor; DIPNECH, diffuse idiopathic pulmonary neuroendocrine cell hyperplasia; NF, nonfunctioning; F, functioning; N/A, not applicable; CS, carcinoid syndrome; CD, Cushing’s disease; ProGRP, pro-gastrin-releasing peptide; CgA, chromogranin A; SD, stable disease; PD, progressive disease.

**Table 2 cancers-15-03282-t002:** Comparison of the circulating markers in lung neuroendocrine neoplasm (LNEN) patients and controls (Mann–Whitney U-Test).

Variable	Groups	Median	IR	*p*
CgA [N: <100 µg/L]	Controls	67.50	40.00–98.00	<0.001
	LNEN	43.43	27.03–81.87	
ProGRP [N: <46 pg/mL]	Controls	6.50	5.50–8.00	<0.0001
	LNEN	136.40	93.90–204.50	

Abbreviations: ProGRP, pro-gastrin-releasing peptide; CgA, chromogranin A; IR, interquartile range.

**Table 3 cancers-15-03282-t003:** Assay positivity in the lung neuroendocrine neoplasm (LNEN) patients.

Essay	Histological LNEN Confirmation (*n* = 290)			
	Total	True Positive	False Negative	Accuracy
ProGRP	290	275	15	95% (275/290)
CgA	290	46	242	29% (46/290)

Abbreviations: ProGRP, pro-gastrin-releasing peptide; CgA, chromogranin A.

**Table 4 cancers-15-03282-t004:** ProGRP and CgA assay metrics in the diagnosis of lung neuroendocrine neoplasm (LNEN) patients.

Variable	AUC (95% CI)	SE	Z Score	*p*	Youden J Index (%)	Cut-Off Value	Sensitivity (%)	Specificity (%)	Accuracy (%)
ProGRP	0.995 (0.99–1.00)	0.003	177.30	<0.001	94.80	32.30 pg/mL	95	100	95
CgA	0.375 (0.31–0.44)	0.035	−3.61	<0.001	15.97	128.25 ug/L	16	100	29

Abbreviations: ProGRP, pro-gastrin-releasing peptide; CgA, chromogranin A; AUC, area under the curve; SE, standard error; CI, confidence interval.

## Data Availability

All data are available upon reasonable request.

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
