# Peer review of "Pro-Gastrin-Releasing Peptide as a Biomarker in Lung Neuroendocrine Neoplasm"

_cancers, 2023, doi:10.3390/cancers15133282_

Round 1
Reviewer 1 Report
This is astudy by Rosiek V et al on proGRP vs CgA in patients with LNENs. This study shows the high sensitivity, specificity and AUC of proGRP in distinguishing patients with LNENs vs control and its superiority compared to CgA. The cohort of TCs and ACs is large.
I have the following minor suggestions:
In simple summary not all the abbreviations are explained eg LNEN.
Again at the simple summary, I can not understand the meaning of the sentence “Huge of the LNEN….”
The first paragraph describing lung cancer could be omitted, the topic is about LNENs.
Line 65: renal/liver impairment
Line 66 ad 67: the abbreviation LNENs should be used.
Table 1: I suggest that the reference interval should be given for proGRP and CgA. Likewise, for Table2.
Line 150, there is double ) after ACs
And the following major suggestions:
The present results are preliminary and this is the reason that I suggest that the levels of proGRP/CgA will be correlated to radiological findings eg tumor load and tumor stage as well as tumor type.
I also suggest that the authors will include follow up of the patients (or some patients) and the response of the biochemical markers to therapy.
The discussion has to be made more attractive.
Minor editing has to be made.
Reviewer 2 Report
AUTHORS
Dear Authors,
I read with interest the manuscript entitled “Pro-Gastrin-Releasing Peptide is Diagnostic of Lung Neuroendocrine Neoplasm”. The authors analyzed the role of pro-GRP as biomarker in Lung neuroendocrine tumors non-SCLC compared to CgA.
The topic is interesting. In fact, even if there are some reports suggesting a role of the proGRP as biomarker in LNEC, there is no consensus and proGRP is not routinely used so far. In this regard the translational perspective seems reasonable. The study is well conducted, results and methods are clear along with the aim of the study. English language is straight and correct. Figure and tables provide a fast comprehension of results and are self-explaining.
However there are some minor issues that should be addressed
1- The Title is too definitive. Even if the results are clear, I recommend to use a milder title as: “Role of proGRP in LNENs” or “ProGRP as biomarker in LNENs”
2- Is not reported if the cohort has been prospectively or retrospectively recruited nor when the blood sample has been collected (first visit, before/after treatment etc). This may affect the proGRP result. Please make more correct the study design and be more specific
3- Strategy: line 73 the sentence “This was a different patient cohort than previously described” is useless
4- It is not clear why SCLC has been excluded from the study. Please include SCLC diagnosis in exclusion criteria if adequate.
5- Lines 87-90 and 149-152 are repetitive. I suggest to include this data only in the result section.
6- I do not make specific comment on the statistical analisys, so I suggest statistical review.
